# A Nonlinear Beam Finite Element with Bending–Torsion Coupling Formulation for Dynamic Analysis with Geometric Nonlinearities

Cesare Patuelli 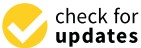, Enrico Cestino *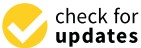 and Giacomo Frulla 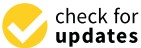

Department of Mechanical and Aerospace Engineering (DIMEAS), Politecnico di Torino, Corso Duca degli Abruzzi 24, 10129 Torino, Italy
* Correspondence: enrico.cestino@polito.it

**Abstract:** Vibration analysis of wing-box structures is a crucial aspect of the aeronautic design to avoid aeroelastic effects during normal flight operations. The deformation of a wing structure can induce nonlinear couplings, causing a different dynamic behavior from the linear counterpart, and nonlinear effects should be considered for more realistic simulations. Moreover, composite materials and aeroelastic tailoring require new simulation tools to include bending–torsion coupling effects. In this research, a beam finite element with bending–torsion coupling formulation is used to investigate the effects of the deflection of beam structures with different aspect ratios. The nonlinear effects are included in the finite element formulation. The geometrical effect is considered, applying a deformation dependent transformation matrix. Stiffness effects are introduced in the stiffness matrix with Hamilton's Principle and a perturbation approach. The results obtained with the beam finite element model are compared with numerical and experimental evidence.

**Keywords:** nonlinear modal analysis; beam finite element; bending–torsion coupling



## 1. Introduction

Aircraft design is moving to increasingly slender and lightweight wings for higher efficiency and lower $CO_2$ emissions. These structures bring new design challenges due to their high flexibility and high deflections under normal loads. The geometric nonlinearities introduced with large deformations require models to predict the effects on the aeroelastic phenomena. The importance of aerodynamic and structural geometrical nonlinearities in the aeroelastic behavior of high-aspect-ratio wings has been established by Patil and Hodges in [1]. Patil et al. [2] have looked at the effect of structural geometric nonlinearities on the flutter behavior of high-aspect-ratio wings, and they presented the changes in the structural and aeroelastic characteristics of a steady-state deflection of a wing. Their study revealed a significant change in the structural frequencies and a significant reduction in the flutter speed.

Detailed coupled computational fluid dynamics and finite element method formulation for aeroelastic analysis have been widely studied [3]. These models can be very sophisticated and generally require a large number of calculations which are not efficient for design and optimization. For this reason, the research moves to low-order aeroelastic models, which consent to reduce the computational cost with similar prediction capabilities if compared to high-order models. A popular approach for nonlinear elasticity consists in geometrically exact beam formulation; an example is given by Hodges [4,5], who presented an intrinsic formulation for nonlinear dynamics of initially curved and twisted anisotropic beams. These models use equivalent beam properties derived from finite element models (FEMs) [6,7]. These formulations have been used in several works for studying very flexible wing structures: Drela [8] used nonlinear beams to develop an integrated model for aerodynamic and structural simulation of flexible aircraft, while Patil [9] presented a theory for

flight dynamic analysis of highly flexible wing configuration, accounting for geometrically nonlinear structural deformations. A more recent class of low-order structural models relies on high-order modal expansions [10,11]. These models require nonlinear static responses of an FEM to identify modal expansion terms. Another model has been presented by Bruni et al. [12], who expanded the partial differential equations for the beam dynamics up to the third order to explore the effects of static deflection, external trim, gust loads, and aerodynamic stall. The solution was obtained with Galerkin's method and with a multi-modal approach. Other models for static and dynamic nonlinear analysis of beam structures exist in the literature and employ different solutions to simulate specific conditions. Some models consider only one-dimensional finite elements as in [13], while other developed models consider all the possible degrees of freedom. Yang et al. [14] developed a six degree of freedom beam element, including material nonlinearity; they described a procedure for nonlinear static analysis which involves piecewise linearization of the response equations and iterations at each incremental step to achieve equilibrium. Surana et al. [15] presented a geometrically nonlinear formulation for a 3D curved beam element using a Lagrangian approach and verified the accuracy and efficiency of the formulation with literature results of nonlinear static analysis. More recently, Jin and Yun [16] developed a three-dimensional beam element for geometrically nonlinear dynamic analysis; the derivation is based on the co-rotational formulation. The model showed good results and can undergo large deflection and rotations, but small strains are assumed.

Low-order beam structural models can be further developed to consider also material couplings and expand the aeroelastic design domain. Anisotropic materials can be used to improve wing box structural performance according to the concept of aeroelastic tailoring [17,18]. The advantages of aeroelastic tailoring are enhanced by orthotropic materials. Bakthavatsalam [19] demonstrated the effect on the flutter speed of aeroelastically tailoring wing and tail surfaces of a closely coupled wing–tail flutter model, and it was shown that tailoring the wing surface produced the largest increase in flutter speed, but tailoring the tail and reducing its stiffness could also produce an increase in flutter speed. Weisshaar [20–22] focused on the use of laminated composites to increase the divergence speeds of swept forward wings. Weisshaar included bending–torsion coupling, defining a stiffness parameter that describes the amount of interaction between the bending curvature and twist rate. This parameter is a function of the orientation and stacking sequence of symmetrical laminate plies with respect to a reference axis along the wing. Composite panels layups can be studied to achieve specific aeroelastic performance and considering also functionally graded materials [23–26] and variable angle tow [27–30]. However, aeroelastic tailoring is not limited to composite materials, as several studies [31–33] have shown that the arrangement of stiffeners can be used to control directional stiffness and bending–torsion coupling.

Beam finite elements are particularly suitable for high-aspect-ratio wing analysis. However, traditional beam elements do not consider the bending–torsion couplings granted by the use of oriented orthotropic materials. Recently, Patuelli et al. [34] presented a beam finite element with bending–torsion coupling formulation (BTCE). The linear finite element was derived with Galerkin's method, while the bending–torsion coupling was obtained with specific shape functions and the hypothesis of constant torsional moment along the beam element. The resulting model was validated with experimental and numerical results [34,35], showing good accuracy for static and dynamic analysis. The scope of the present research work is to develop a procedure to perform dynamic analysis in the presence of geometric nonlinearities. Cestino et al. [36] studied the flutter instability of high-aspect-ratio wings and considered the phenomenon as the sum of two effects, the geometrical effect (GE) given by the deformed geometry and the stiffness effect (SE), which is the effect caused by the loads at the equilibrium condition on the differential stiffness matrix. They demonstrated that the GE is the main contribution in the nonlinear dynamic analysis of slender structures and that the results of the flutter analysis are verified by experimental evidence either when considering only GE or when accounting for SE.

This research work presents the derivation of a nonlinear beam finite element with bending–torsion coupling formulation (BTCE-NL), which considers both GE and SE. The element stiffness matrix is derived considering the nonlinear terms through the perturbation of a known equilibrium configuration. Moreover, an approach that accounts only for the GE (BTCE-GE) has been developed considering a deformed equilibrium dependent transformation matrix to orient the BTCE. The derived models have been validated with several experimental modal analyses performed with Laser Doppler Vibrometer (LDV). The experimental data gathered have been used to validate the characteristic frequencies and the mode shape predicted with the new models, but they also consented to understand at which level of deflection the dynamic linear analysis is no longer suitable for mode shape prediction, and advanced models are needed. The experimental tests considered a beam structure with a coupling coefficient equal to zero to avoid flapwise vibrations. The results revealed that both models can predict characteristic frequencies and modes with good precision for moderate-to-large deflections and that the BTCE-GE can be sufficient to analyze slender structures with moderate initial deflections. The models has been tested also for a composite beam-box structure with a circumferentially asymmetric stiffness (CAS) configuration described in [37]. The experimental modal analysis of a beam structure with bending–torsion couplings involves both edgewise and flapwise coupled modes and would require more sophisticated equipment; for this reason, the BTCE models results have been compared with the results of a SHELL FE model solved with NASTRAN SOL106. The methods adopted showed good correlation with the SHELL FE model for moderate deformations. The models presented in this research can be used to perform optimization cycles with low computational costs and find layup configurations able to mitigate the deformation-induced nonlinear effects.

## 2. Model Derivations

In this section, the BTCE derived in [34] is used to develop two procedures for modal analysis of predeformed structures. The first procedure considers only the geometrical effects given by the deformed configuration, while the second procedure uses a perturbation approach to include nonlinear effects in the beam element stiffness matrix. The generic beam finite element is represented in Figure 1, each node presenting three translations and three rotations. Out-of-plane bending involves vertical translation $w$ and rotation around the $z$ axis $\theta_z$, while in-plane bending involves horizontal translation $v$ and rotation around the $y$ axis $\theta_y$. The torsion of the beam is described by the rotation around the beam axis $\varphi$ and the extension of the beam by translation $u$.

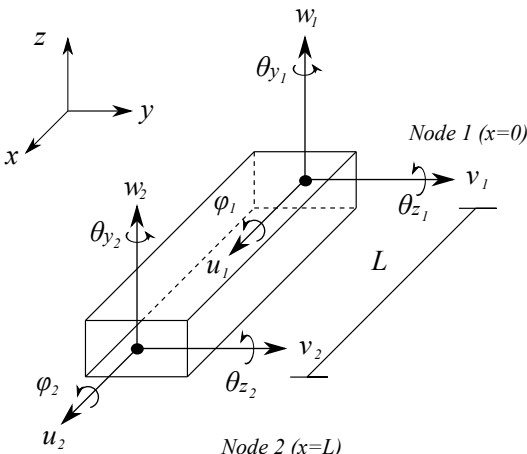

**Figure 1.** Beam element reference system.

### 2.1. BTCE-GE

The BTCE stiffness and mass matrices $[K_{el}]$ and $[M_{el}]$ reported in Equations (A1) and (A3) can be obtained through Galerkin's method following the procedure described in [34]. The

BTCE formulation allows to consider the bending–torsion coupling given by oriented fibers or stiffeners for an inextensible slender box beam in a CAS configuration. This coupling is obtained through the shape functions reported in Table A1, derived with the hypothesis of constant torsional moment along the beam element. To perform a dynamic analysis of a predeformed structure considering only the geometrical effects, the beam can be discretized in finite elements with the position and orientation given by a known equilibrium configuration. This can be achieved with the rotation of the mass and stiffness matrices with an opportune transformation matrix $[T]$ defined in Equation (1):

$$[T] = \begin{bmatrix} T_{11} & T_{12} & T_{13} \\ T_{21} & T_{22} & T_{23} \\ T_{31} & T_{32} & T_{33} \end{bmatrix} \tag{1}$$

The members $T_{ij}$ for an inextensible beam are reported in Equation (2) according to the derivation presented in [38]. The $T_{ij}$ are obtained with a Taylor's expansion truncated at the second order, which introduces the hypothesis of moderate-to-large deformations with deflections between 10% and 15% of the beam length:

$$\begin{cases} T_{11} = 1 - \dfrac{1}{2}v'^2 - \dfrac{1}{2}w'^2 \\ T_{12} = v' \\ T_{13} = w' \\ T_{21} = -v' - w'\varphi \\ T_{22} = 1 - \dfrac{1}{2}v'^2 - \dfrac{1}{2}\varphi^2 \\ T_{23} = \varphi \\ T_{31} = -w' + \varphi v' \\ T_{32} = -\varphi - v'w' \\ T_{33} = 1 - \dfrac{1}{2}w'^2 - \dfrac{1}{2}\varphi^2 \end{cases} \tag{2}$$

$v$, $w$, and $\varphi$ are the displacement variables function of the coordinate $x$. The known deformed configuration can be denoted with $v_0$, $w_0$, and $\varphi_0$. For the two-node finite element represented in Figure 1, the equilibrium deformation can be expressed as the product of shape functions times the nodal degrees of freedom of the element as represented in Equation (3), the suffix 0 denoting the equilibrium value of the degree of freedom:

$$\begin{cases} w_0(x) = N_{w1}(x)w_{01} + N_{w2}(x)\theta_{y01} + N_{w3}(x)w_{02} + N_{w4}(x)\theta_{y02} \\ v_0(x) = N_{v1}(x)v_{01} + N_{v2}(x)\theta_{z01} + N_{v3}(x)v_{02} + N_{v4}(x)\theta_{z02} \\ \varphi_0(x) = N_{\varphi 1}(x)\varphi_{01} + N_{\varphi 2}(x)w_{01} + N_{\varphi 3}(x)\theta_{y01} + N_{\varphi 4}(x)\varphi_{02} + N_{\varphi 5}(x)w_{02} + N_{\varphi 6}(x)\theta_{y02} \end{cases} \tag{3}$$

Substituting Equation (3) into Equation (2), the transformation matrix is obtained for each element of the structure. However, the members $T_{ij}$ vary along the beam element length. The orientation of the element can be obtained evaluating the $T_{ij}$ at the first node of the beam element. This procedure introduces the hypothesis that the deformation along the element are negligible, and it can be considered straight. Once the matrix $[T]$ is obtained, the finite element mass and stiffness matrices in local coordinates $[K_{el}]$ and $[M_{el}]$ can be rotated according to the orientation of the deformed structure with Equations (4) and (5). The oriented element can be assembled to solve the eigenvalue problem (6), computing the corresponding eigenvectors solution of Equation (7) with the global stiffness and mass matrices $[K_g]$ and $[M_g]$:

$$[K_g] = [T]^T[K_{el}][T] \tag{4}$$

$$[M_g] = [T]^T[M_{el}][T] \tag{5}$$

$$det\left([K_g] - \omega_n^2[M_g]\right) = 0 \tag{6}$$

$$([K_g] - \omega_n^2[M_g])\phi_n = 0 \tag{7}$$

*2.2. BTCE-NL*

The second procedure presented in this research work takes into account the stiffness effect of an equilibrium deformed configuration. Considering a uniform straight orthotropic inextensible beam, the Cartesian coordinate system XYZ describes the undeformed geometry, and the Cartesian system $\xi\eta\zeta$ describes the deformed geometry (Figure 2). The derivation uses Hamilton's Principle reported in Equation (8):

$$\int_0^t (\partial T - \partial \pi + \partial W_{nc})dt = 0 \tag{8}$$

where $\partial T$ is the kinetic energy and $\partial W_{nc}$ are the nonconservative terms. The variation of the elastic energy $\partial \pi$ can be written as in Equation (8) according to [38]:

$$\partial \pi = \int_0^L (M_1 \partial \rho_1 + M_2 \partial \rho_2 + M_3 \partial \rho_3)ds = 0 \tag{9}$$

with the moment resultants:

$$\begin{cases} M_1 = GJ\rho_1 + K\rho_2 \\ M_2 = EI_2\rho_2 + K\rho_1 \\ M_3 = EI_3\rho_3 \end{cases} \tag{10}$$

The curvatures $\rho_1$, $\rho_2$ and $\rho_3$ are obtained from the transformation matrix [T] according to [38]:

$$\begin{cases} \rho_1 = \varphi' + v''w' \\ \rho_2 = -w'' + v''\varphi \\ \rho_3 = v'' + w''\varphi \end{cases} \tag{11}$$

and the assumption that allows to obtain linear equations is to consider the displacement variables as the sum of an equilibrium term denoted with the suffix 0 and a perturbation term:

$$\begin{cases} \varphi = \varphi_0 + \tilde{\varphi} \\ w = w_0 + \tilde{w} \\ v = v_0 + \tilde{v} \end{cases} \tag{12}$$

where substituting Equation (12) in Equation (11) and neglecting the equilibrium terms, Equation (11) becomes:

$$\begin{cases} \rho_1 = \tilde{\varphi}' + v_0''\tilde{w}' + w_0'\tilde{v}'' \\ \rho_2 = -\tilde{w}'' + v_0''\tilde{\varphi} + \varphi_0\tilde{v}'' \\ \rho_3 = \tilde{v}'' + w_0''\tilde{\varphi} + \varphi_0\tilde{w}'' \end{cases} \tag{13}$$

and the differential of the curvatures can be written as:

$$\begin{cases} \partial\rho_1 = \partial\tilde{\varphi}' + v_0''\partial\tilde{w}' + w_0'\partial\tilde{v}'' \\ \partial\rho_2 = \partial(-\tilde{w})'' + v_0''\partial\tilde{\varphi} + \varphi_0\partial\tilde{v}'' \\ \partial\rho_3 = \partial\tilde{v}'' + w_0''\partial\tilde{\varphi} + \varphi_0\partial\tilde{w}'' \end{cases} \tag{14}$$

With these considerations, is it possible to substitute Equations (13) and (14) into Equation (9). Considering only the perturbation terms, it is possible to obtain Equation (15). The three members of Equation (9) are presented separately for the sake of clarity:

$$\begin{cases} M_1 \partial \rho_1 = [GJ(\tilde{\varphi}' + v_0''\tilde{w}' + w_0'\tilde{v}'') + K(-\tilde{w}' + v_0''\tilde{\varphi} + \varphi_0\tilde{v}'')]\partial \rho_1 \\ M_2 \partial \rho_2 = [EI_2(-\tilde{w}'' + v_0''\tilde{\varphi}' + \varphi_0\tilde{v}'') + K(\tilde{\varphi}' + v_0''\tilde{w}' + w_0'\tilde{v}'')]\partial \rho_2 \\ M_3 \partial \rho_3 = [EI_3(\tilde{v}'' + w_0''\tilde{\varphi} + \varphi_0\tilde{w}'')]\partial \rho_3 \end{cases} \tag{15}$$

Equation (15) can be written in matrix form as

$$\begin{cases} M_1 \partial \rho_1 = \{\partial \tilde{d}\} \{ \begin{matrix} 1 & 0 & w_0' & v_0'' & 0 \end{matrix} \}^T \{ \begin{matrix} GJ & K & GJw_0' + K\varphi_0 & GJv_0'' & Kv_0'' \end{matrix} \} \{\tilde{d}\}^T \\ M_2 \partial \rho_2 = \{\partial \tilde{d}\} \{ \begin{matrix} 0 & 1 & \varphi_0 & 0 & v_0'' \end{matrix} \}^T \{ \begin{matrix} K & EI_2 & EI_2\varphi_0 + Kw_0' & Kv_0'' & EI_2v_0'' \end{matrix} \} \{\tilde{d}\}^T \\ M_3 \partial \rho_3 = \{\partial \tilde{d}\} \{ \begin{matrix} 0 & -\varphi_0 & 1 & 0 & w_0'' \end{matrix} \}^T \{ \begin{matrix} 0 & -EI_3\varphi_0 & EI_3 & 0 & EI_3w_0'' \end{matrix} \} \{\tilde{d}\}^T \end{cases} \tag{16}$$

with $\{\partial \tilde{d}\}$ and $\{\tilde{d}\}$ defined as in Equations (17) and (18)

$$\{\partial \tilde{d}\} = \{ \begin{matrix} \partial \tilde{\varphi}' & \partial(-\tilde{w}'') & \partial \tilde{v}'' & \partial \tilde{w}' & \partial \tilde{\varphi} \end{matrix} \} \tag{17}$$

$$\{\tilde{d}\} = \{ \begin{matrix} \tilde{\varphi}' & (-\tilde{w}'') & \tilde{v}'' & \tilde{w}' & \tilde{\varphi} \end{matrix} \} \tag{18}$$

and the vectors containing only equilibrium terms can be multiplied, obtaining:

$$\begin{cases} M_1 \partial \rho_1 = \{\partial \tilde{d}\} \begin{bmatrix} GJ & K & GJw_0' + K\varphi_0 & GJv_0'' & Kv_0'' \\ 0 & 0 & 0 & 0 & 0 \\ GJw_0' & Kw_0' & w_0'(GJw_0' + K\varphi_0) & GJv_0''w_0' & Kv_0''w_0' \\ GJv_0'' & Kv_0'' & v_0''(GJw_0' + K\varphi_0) & GJ(v_0'')^2 & K(v_0'')^2 \\ 0 & 0 & 0 & 0 & 0 \end{bmatrix} \{\tilde{d}\}^T \\ \\ M_2 \partial \rho_2 = \{\partial \tilde{d}\} \begin{bmatrix} 0 & 0 & 0 & 0 & 0 \\ K & EI_2 & EI_2\varphi_0 + Kw_0' & Kv_0'' & EI_2v_0'' \\ K\varphi_0 & EI_2\varphi_0 & \varphi_0(EI_2\varphi_0 + Kw_0') & Kv_0''\varphi_0 & EI_2v_0''\varphi_0 \\ 0 & 0 & 0 & 0 & 0 \\ Kv_0'' & EI_2v_0'' & v_0''(EI_2\varphi_0 + Kw_0') & K(v_0'')^2 & EI_2(v_0'')^2 \end{bmatrix} \{\tilde{d}\}^T \\ \\ M_3 \partial \rho_3 = \{\partial \tilde{d}\} \begin{bmatrix} 0 & 0 & 0 & 0 & 0 \\ 0 & -EI_3\varphi_0^2 & -EI_3\varphi_0 & 0 & -EI_3w_0''\varphi_0 \\ 0 & -EI_3\varphi_0 & EI_3 & 0 & EI_3w_0'' \\ 0 & 0 & 0 & 0 & 0 \\ 0 & -EI_3\varphi_0w_0'' & EI_3w_0'' & 0 & EI_3(w_0'')^2 \end{bmatrix} \{\tilde{d}\}^T \end{cases} \tag{19}$$

Equation (9) can be rewritten in the matrix form as:

$$\partial \pi = \int_0^L \{\partial \tilde{d}\}[\tilde{C}]\{\tilde{d}\}^T ds = 0 \tag{20}$$

The matrix $[\tilde{C}]$ is obtained with the sum of the matrices in Equation (19):

$$[\tilde{C}] = \begin{bmatrix} GJ & K & GJw_0' + K\varphi_0 & GJv_0'' & Kv_0'' \\ & EI_2 + EI_3\varphi_0^2 & EI_2\varphi_0 + Kw_0' - EI_3\varphi_0 & Kv_0'' & EI_2v_0'' - EI_3w_0''\varphi_0 \\ & & EI_3 + GJ(w_0')^2 + K\varphi_0w_0' + EI_2(\varphi_0)^2 + Kw_0'\varphi_0 & GJv_0''w_0' + Kv_0''\varphi_0 & EI_2v_0''\varphi_0 + EI_3w_0'' + Kv_0''w_0' \\ & & & GJ(v_0'')^2 & K(v_0'')^2 \\ & & & & EI_2(v_0'')^2 + EI_3(w_0'')^2 \end{bmatrix} \tag{21}$$

The displacement variables can be expressed as a set of space-dependent shape functions $[N(x)]$, which multiplies the time-dependent nodal degrees of freedom $\{\tilde{q}\}$. In this

case, the shape functions used are the ones reported in Appendix A, derived in [34], and include the bending–torsion coupling formulation:

$$\tilde{d} = [N(x)]\{\tilde{q}(t)\}^T \tag{22}$$

Substituting the shape functions into Equation (20), it can be rewritten as:

$$\partial \pi = \partial \{\tilde{q}\} \left( \int_0^L [N(x)]^T [\tilde{C}][N(x)]ds \right) \{\tilde{q}\}^T = 0 \tag{23}$$

where the nonlinear stiffness matrix is expressed as:

$$[\tilde{K}] = \left( \int_0^L [N(x)]^T [\tilde{C}][N(x)]ds \right) \tag{24}$$

A known equilibrium configuration can be used to compute the nonlinear stiffness matrix $[\tilde{K}]$ and solve eigenvalue problem (6), computing the corresponding eigenvectors solution of Equation (7). The mass matrix $[M]$ can be assembled using the linear BTCE formulation since the effect of the nonlinear terms is negligible.

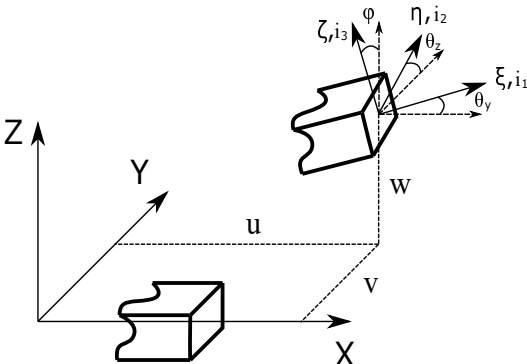

**Figure 2.** Beam reference system.

## 3. Experimental Validation for Isotropic Beam

The models here derived have been validated through experimental modal analysis. The tests were performed on a rectangular section aluminum 6060 beam with dimensions $L = 3000$ mm, $b = 40$ mm, and $h = 8$ mm (Figure 3) and the mechanical properties listed in Table 1. The beam was clamped in four different positions to gather data of four cases, respectively, with useful length $L_1 = 1000$ mm, $L_2 = 1500$ mm, $L_3 = 2000$ mm, and $L_4 = 2500$ mm (Figure 4). We defined the ratio $\lambda = \mu/L$ with $\mu$ equal to the tip deflection; one of the scopes of the experimental test is to understand at which level of $\lambda$ the geometric nonlinear effects have an influence on the beam mode shapes and characteristic frequencies, determining the need of a nonlinear modal analysis. The other objective for the experimental testing is to verify the accuracy of the presented models in predicting characteristic frequencies and mode shapes.

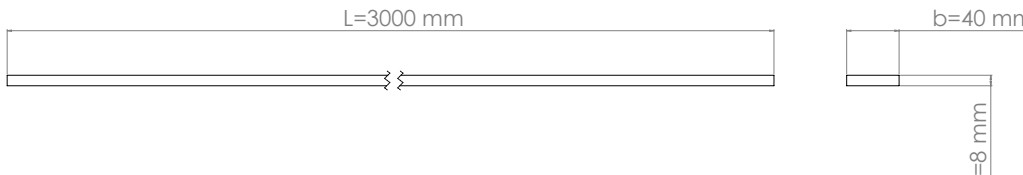

**Figure 3.** Beam dimensions.

**Table 1.** Aluminum 6060 mechanical properties.

| Property | Value |
|---|---|
| E | 61,000 [MPa] |
| $\nu$ | 0.3 |
| $\rho$ | 2675 Kg/m$^3$ |

The beam was investigated with four experimental tests with similar equipment. The beam was clamped between two steel blocks to guarantee a rigid constraint (Figure 5) at the first section of the beam. Ten polylactic acid (PLA) targets (Figure 5) were placed along the beam to gather data at ten equidistant stations. The number of targets was limited to ten units to keep the additional weight on the beam negligible. More targets can be added to improve the acquisition resolution, but the mass and inertia must be considered and can alter the nonlinear effects observed. Each target presented two vertical parts for acquisition, where a squared piece of reflective tape was positioned to improve the surface reflectiveness.

The acquisition was performed with a Polytec PSV-500 Laser Doppler Vibrometer (LDV) system, while the excitation was obtained with an electrodynamic shaker K200xE01. The shaker was placed at 550 mm from the constraint, perpendicular to the beam axis as represented in Figure 5. The objective was to excite only the edgewise degree of freedom because edgewise and torsional characteristic frequencies are the most affected by flapwise deflection; moreover, the torsional modes should be visible only when the nonlinear coupling effect becomes important according to [1].

The experimental validation considers the case where the coupling term K is equal to 0. This allows to reduce the number of variables and keep the interpretation of the results straightforward. Moreover, the term K induces bending–torsion deformations, which means that a mode shape which involves the torsional degree of freedom determines also the flapwise displacements, which need a 3D LDV system to be detected.

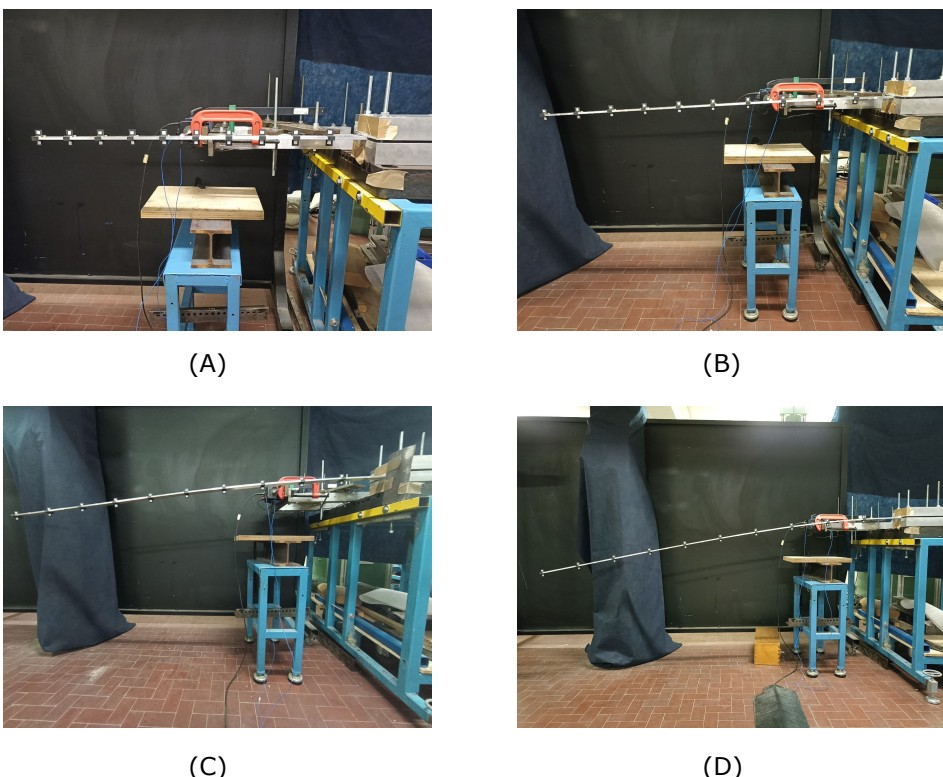

(A)

(B)

(C)

(D)

**Figure 4.** Experimental setup: *L* = 1000 mm (**A**), *L* = 1500 mm (**B**), *L* = 2000 mm (**C**), *L* = 2500 mm (**D**).

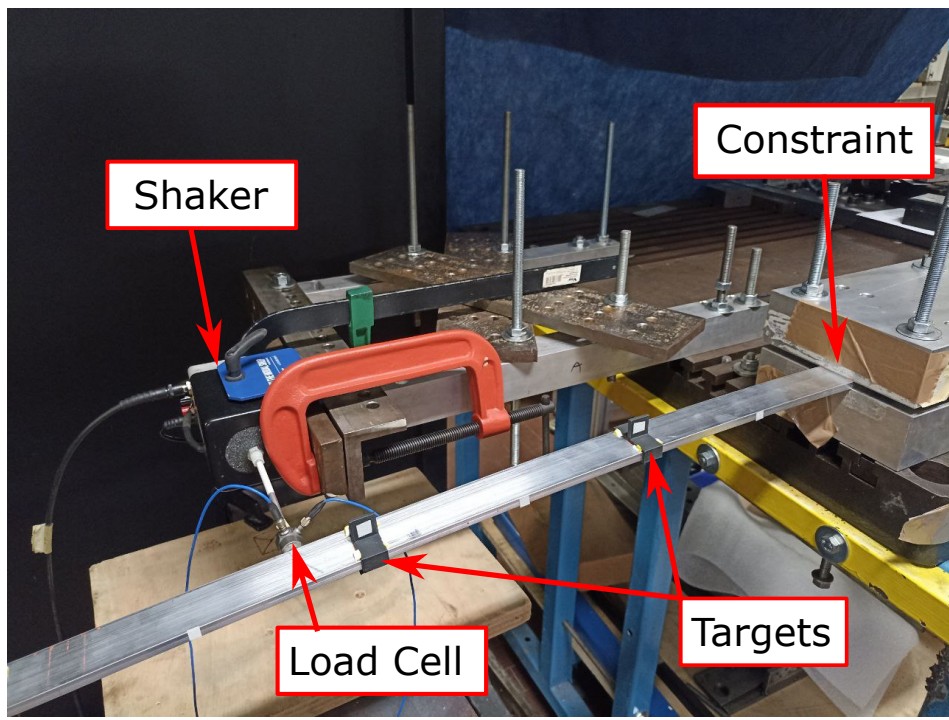

**Figure 5.** Experimental Setup.

Four numerical models were defined for experimental result comparison. Two models were implemented with MATLAB and used the BTCE finite element. One used the linear formulation derived in [34], accounting for the geometrical effect (BTCE-GE), while the second used the nonlinear BTCE (BTCE-NL). Two additional models were defined in PATRAN and solved in NASTRAN starting with an undeformed configuration; both used SHELL elements to describe the beam geometry, but one was solved with SOL103 and the second one was solved with the nonlinear solution SOL106. The linear modal analysis was performed on the undeformed configuration to obtain linear mode shapes and frequencies for the result comparison. The choice behind the use of SHELL finite elements is the possibility to add bending–torsion coupling terms, which is possible for the BTCE but not for conventional beam elements. The BTCE models were obtained by assembling 10 elements which represent the 10 segments described by the targets positioned on the experimental beam. The first node was constrained, imposing the translations and the rotations equal to 0. For the BTCE-GE model, the modal analysis was performed using the linear stiffness matrix rotated with the equilibrium configuration-dependent transformation matrix $[T]$ described in the second section of this work. The BTCE-NL model used the stiffness matrix derived in the second section, which depends on the equilibrium static deformation. The mass of each element was lumped at the nodes, and a linear static analysis determined the equilibrium deformation used to complete the element stiffness matrix and perform the nonlinear modal analysis. Alternatively, the deformed configuration can be obtained with a nonlinear static analysis performed with NASTRAN. The linear static analysis for a vertical load does not present edgewise displacements or rotation, while the nonlinear static analysis have a small in-plane component $v_0$ and $\theta_{z0}$, which can be considered negligible. The SHELL elements was created with 10 mm QUAD4 elements and then solved with SOL103 for the linear modal analysis in the undeformed configuration as reference. The model was completed with an inertial load to perform also the nonlinear modal analysis SOL106 that accounts for the preload. The numerical models results were compared with the experimental results in terms of mode shapes and characteristic frequencies. The linear analysis was performed to understand at which level of deformation it becomes unreliable and a nonlinear formulation becomes needed.

## 4. Numerical Models Comparison for Composite Beam

A numerical comparison was performed for a case with bending–torsion coupling; the reference structure is a box beam structure with a circumferentially asymmetric stiffness (CAS) laminated composite configuration. The structure was described in [37]; the section is represented in Figure 6. The beam was obtained with a unidirectional T700 carbon–epoxy layer bonded onto wooden spars with fibers oriented at 24°. The structural box had the following dimensions: length, $L$ = 522 mm; width, w = 20 mm; height, h = 2.8 mm; upper and lower panel thicknesses, t = 0.2 mm; mass per unit length, m = $1.095 \times 10^{-5}$ kg/mm; and torsional unit inertia, Ip = $4.75 \times 10^{-4}$ kg/mm. The mechanical properties of the material are listed in Table 2.

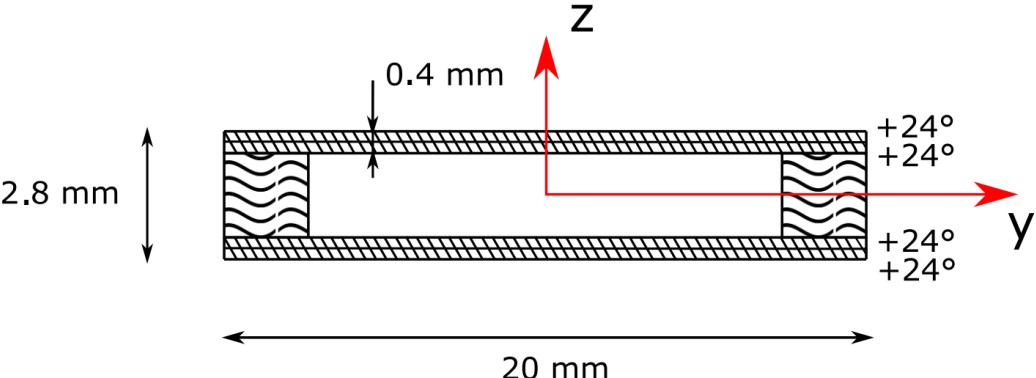

**Figure 6.** Composite box beam section.

**Table 2.** Cantilever composite beam material and properties.

| T700 Property | Value | Wood Property | Value |
|---|---|---|---|
| $E_{11}$ | 118.4 GPa | $E_{11}$ | 16.6 GPa |
| $E_{22}$ | 8.7 GPa | $E_{22}$ | 8 GPa |
| $G_{12} = G_{13}$ | 3.4 GPa | $G_{12} = G_{13}$ | 3.4 GPa |
| $v_{12} = v_{13}$ | 0.31 | $v_{12} = v_{13}$ | 0.31 |

The reference model was defined in PATRAN with SHELL elements (Figure 7), while a beam model with the formulation presented in this work was obtained assembling 10 elements. The load condition chosen for the numerical comparison is a concentrated tip load. The load was incremented to reach different deformation levels and observe the limits of validity of the presented model. The deformation was evaluated with a nonlinear static analysis, then a nonlinear modal analysis was performed for each load case, and the numerical results were compared in terms of characteristic frequencies. The deformed configuration used to orient and compute the nonlinear beam finite element was retrieved from the nonlinear static analysis performed with NASTRAN. The first eight characteristic frequencies were computed for the two FE models and normalized with the value obtained with a linear modal analysis of the undeformed configuration. The normalized frequencies were compared for each mode at different deformation levels.

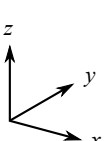

**Figure 7.** Composite box beam SHELL model.

## 5. Results and Discussion

This section presents the experimental evidences collected and compared with the numerical models. The dynamic behavior of the nonlinear BTCE was compared in terms of predicted natural frequencies and mode shapes. The accuracy of the natural frequencies was evaluated in terms of relative difference. The similarity between the FE models and the experimental mode shapes was evaluated with the Modal Assurance Criterion (MAC). MAC is a statistical indicator used to quantifying the similarity between two sets of mode shapes, where a value equal to 1 indicates complete similarity, while 0 indicates no correlation between the modes investigated [39,40]. Equation (25) was applied to the experimental and numerical mode shapes computed to obtain the MAC matrices. The mode sets of the experimental mode shapes were compared with themselves computing the Auto MAC, allowing to verify the existence of similarities between different mode shapes and thus the presence of couplings between the degrees of freedom. The couplings, if present, should show the same pattern for experimental and nonlinear numerical modes. When the structure does not present couplings, the expected matrices for the experimental and numerical linear and nonlinear modes should be diagonal:

$$MAC_{ij} = \frac{|\boldsymbol{\Phi}_A^{iT}\boldsymbol{\Phi}_B^{j}|^2}{\left(\boldsymbol{\Phi}_A^{iT}\boldsymbol{\Phi}_A^{j}\right)\left(\boldsymbol{\Phi}_B^{iT}\boldsymbol{\Phi}_B^{j}\right)} \tag{25}$$

### 5.1. Static Analysis Results

The nonlinear finite element derived depends on the equilibrium deformation under static load. The deformation can be obtained through linear or nonlinear static analysis. In this research, a linear static analysis is used to determine the initial equilibrium deformation for the load cases considered during the experimental tests on the isotropic beam. The results of the deflection at the tip were compared with the result of a SHELL model of the beam solved with SOL106 and experimental results, and the accuracy was evaluated computing the relative difference between numerical and experimental results. The comparison is reported in Table 3 with the relative difference for each result.

**Table 3.** Comparison of experimental deflection measured at the tip with linear and nonlinear static analysis results. $\mu$ denotes the tip deflection, while L denotes the beam length.

| Beam Length [mm] | Experimental [mm] | BTCE [mm] | SHELL SOL106 [mm] | $\mu/L\%$ |
|:---:|:---:|:---:|:---:|:---:|
| 1000 | 10 | 10 | 10 | 1.1% |
|  |  | 0% | 0% |  |
| 1500 | 54 | 51 | 51 | 3.7% |
|  |  | 5.5% | 5.5% |  |
| 2000 | 166 | 162 | 160 | 8.3% |
|  |  | 2.4% | 3.6% |  |
| 2500 | 367 | 395 | 386 | 14.68% |
|  |  | 7.6% | 5.2% |  |

### 5.2. Experimental Modal Analysis Results

The Frequency Response Functions (FRFs) obtained through experimental modal analysis are reported in Figure 8. The experimental and numerical results for the characteristic frequencies are reported in Table 4. For a beam length equal to 1000 mm, the first torsional mode was not detected, while for a length of 1500 mm, the torsional mode was detected, but the peak was significantly smaller than the others. This confirms that the coupling between edgewise bending and torsion is weak for deformations below 5%. On the other hand, for bigger deformations, the excitation of the edgewise degree of freedom provoked also the detection of the first torsional mode coupled with the edgewise bending mode.

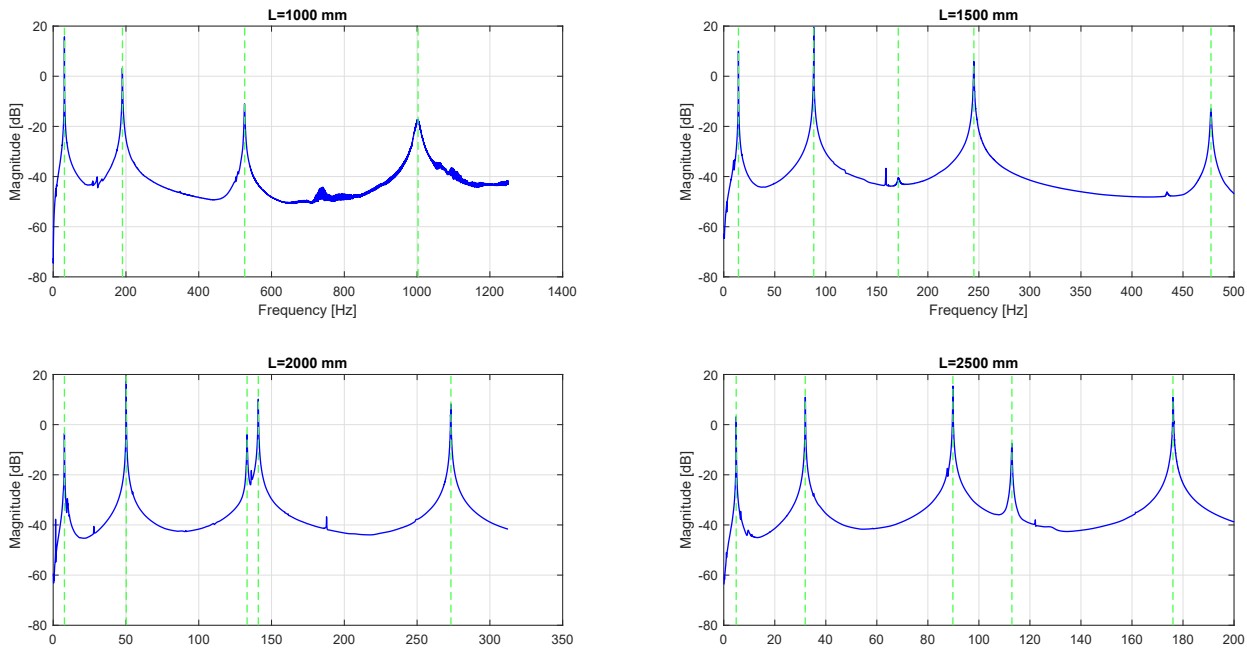

**Figure 8.** Experimental FRF.

The frequencies reported in Table 4 show good accordance with the predicted values and the experimental results. In this case, both linear and nonlinear models can be use to determine the characteristic frequencies of the structures. The relative difference between the predicted and observed frequencies, reported within parentheses in Table 4, is generally below 5% with some exception compatible with the approximations introduced with the derivation of the BTCE models. Moreover, the difference between the BTCE-GE model and the BTCE-NL model are minimal, confirming the findings reported in [36,41] concerning the major contribution of the geometrical effect in this class of analysis.

**Table 4.** Experimental and numerical natural frequencies results comparison for the isotropic beam.

| | L = 1000 mm | | | | | |
|---|---|---|---|---|---|---|
| Mode | Exp | Analytical LIN [Hz] | SOL103 [Hz] | BTCE-GE [Hz] | BTCE-NL [Hz] | SOL106 [Hz] |
| 1E | 30.86 | 30.86 | 30.84 | 30.85 | 30.85 | 30.83 |
| | | (0%) | (0.06%) | (0.03%) | (0.03%) | (0.03%) |
| 2E | 190.14 | 193.37 | 191.93 | 193.36 | 193.34 | 191.8 |
| | | (1.7%) | (0.94%) | (1.67%) | (1.68%) | (0.87%) |
| 3E | 525.68 | 541.45 | 531.55 | 541.57 | 541.56 | 530.87 |
| | | (3%) | (1.12%) | (3.02%) | (3.02%) | (0.98%) |
| 4E | 1001.56 | 1061.02 | 1025.7 | 1062.01 | 1062.00 | 1023.4 |
| | | (5.94%) | (2.41%) | (6.03%) | (6.03%) | (2.18%) |
| 1T | / | 271.34 | 279.95 | 271.67 | 271.71 | 264.02 |
| | L = 1500 mm | | | | | |
| Mode | Exp | Analytical LIN [Hz] | SOL103 [Hz] | BTCE-GE [Hz] | BTCE-NL [Hz] | SOL106 [Hz] |
| 1E | 14.38 | 13.71 | 13.71 | 13.68 | 13.66 | 13.68 |
| | | (4.66%) | (4.66%) | (4.87%) | (5.01%) | (4.87%) |
| 2E | 88.20 | 85.94 | 85.67 | 85.87 | 85.82 | 85.55 |
| | | (2.56%) | (2.89%) | (2.64%) | (2.7%) | (3.0%) |
| 3E | 245.08 | 240.64 | 238.69 | 240.67 | 240.63 | 238.5 |
| | | (1.81%) | (2.61%) | (1.80%) | (1.82%) | (2.68%) |
| 4E | 477.42 | 471.56 | 464.43 | 471.85 | 471.86 | 463.79 |
| | | (1.23%) | (2.72%) | (1.16%) | (1.16%) | (2.85%) |
| 1T | 171.02 | 180.89 | 186.14 | 181.43 | 181.61 | 176.0 |
| | | (5.75%) | (8.84%) | (6.08%) | (6.19%) | (2.91%) |
| | L = 2000 mm | | | | | |
| Mode | Exp | Analytical LIN [Hz] | SOL103 [Hz] | BTCE-GE [Hz] | BTCE-NL [Hz] | SOL106 [Hz] |
| 1E | 7.71 | 7.71 | 7.71 | 7.62 | 7.56 | 7.60 |
| | | (0%) | (0%) | (1.17%) | (1.95%) | (1.43%) |
| 2E | 50.20 | 48.34 | 53.12 | 48.13 | 48.02 | 47.96 |
| | | (3.71%) | (5.82%) | (4.12%) | (4.34%) | (4.46%) |
| 3E | 133.20 | 135.36 | 134.75 | 133.67 | 134.12 | 131.9 |
| | | (1.62%) | (1.15%) | (0.35%) | (0.69%) | (0.98%) |
| 4E | 273.24 | 265.25 | 262.99 | 265.15 | 265.15 | 262.49 |
| | | (2.92%) | (3.75%) | (2.96%) | (2.96%) | (3.93%) |
| 1T | 140.92 | 135.67 | 139.42 | 138.89 | 138.99 | 135.96 |
| | | (3.72%) | (1.06%) | (1.44%) | (1.37%) | (3.52%) |
| | L = 2500 mm | | | | | |
| Mode | Exp | Analytical LIN [Hz] | SOL103 [Hz] | BTCE-GE [Hz] | BTCE-NL [Hz] | SOL106 [Hz] |
| 1E | 4.86 | 4.94 | 4.94 | 4.71 | 4.61 | 4.68 |
| | | (1.65%) | (1.65%) | (3.18%) | (5.42%) | (3.7%) |
| 2E | 31.94 | 30.94 | 30.94 | 30.44 | 30.27 | 30.29 |
| | | (3.13%) | (3.13%) | (3.33%) | (5.73%) | (5.44%) |
| 3E | 89.81 | 86.63 | 86.39 | 85.76 | 85.82 | 85.54 |
| | | (3.54%) | (3.81%) | (4.51%) | (4.44%) | (4.86%) |
| 4E | 176.06 | 169.76 | 168.84 | 169.16 | 169.19 | 168.04 |
| | | (3.58%) | (4.1%) | (3.92%) | (3.90%) | (4.56%) |
| 1T | 112.94 | 108.53 | 111.44 | 113.62 | 114.71 | 110.97 |
| | | (3.9%) | (1.33%) | (0.60%) | (1.57%) | (1.74%) |

The experimental Auto MAC matrices are reported in Figures 9A–12A. The mode order is based on the frequency value, from the mode with the lowest frequency to the one with the highest. With this convention, for L = 1000 mm and L = 1500 mm, the torsional mode occupies the third position, while for the other cases, it is placed in the fourth position. The remaining modes represent the edgewise modes. In Figure 9, it is possible to notice the absence of the torsional mode. As already stated, when nonlinear effects are not present, edgewise displacement and torsion are not coupled, thus exciting the edgewise displacement; the torsional mode cannot be observed. On the other hand, the numerical modes predict five uncoupled modes as expected.

For L = 1500 mm, the experimental Auto MAC matrix reported in Figure 10A reveals a certain level of coupling between the torsional mode and the second and third edgewise modes. This coupling is not detected by the linear FE models, while it is present in the nonlinear FE model MAC matrices. The experimental and nonlinear FE model MAC matrices present some differences in the out of the diagonal values. In this case, the torsional

mode presents a small peak in the FRF because the nonlinear effect is present but not very relevant, with 3.7% deflection. Moreover, the number of targets is relatively low and can cause some discrepancies. However, it is possible to conclude that for $L$ = 1500 mm, the nonlinear effect is present and can be predicted with the nonlinear BTCE models, but linear modal analysis can be a reasonable approximation for this level of deflection. The Auto MAC matrices for $L$ = 2000 mm and $L$ = 2500 mm are reported in Figures 11A and 12A. The pattern given by the experimental results is correctly predicted by the nonlinear models; moreover, these cases highlight the lack of accuracy obtained when linear models are considered for the modal analysis of structures with moderate deformations.

Figures 9–12 present also the comparison between the experimental modes and the numerical modes calculated with linear and nonlinear FE models. Ideally, if the numerical modes are coincident with the experimental modes, the MAC matrices should be identical to the Auto MAC experimental matrices. In general, it is possible to affirm that the mode shape predicted with the nonlinear BTCE model is in good accordance with the experimental modes; moreover, they are confirmed by the results of the SHELL FE model solved with NASTRAN SOL106. In the first case ($L$ = 1000 mm), the torsional mode was not detected, and for this reason, the comparison with numerical counterpart is not reported in Figure 9. The MAC matrices for the beam with $L$ = 1500 mm reveal a high similarity with the experimental results when nonlinear modal analysis is used, while the similarity is considerably lower when the nonlinear effects are not considered. This is even more evident for $L$ = 2000 mm and $L$ = 2500 mm. The fourth case presents a relatively low similarity for the fourth mode (Figure 12), which corresponds to the third edgewise mode coupled with the torsional mode. This is probably connected to the resolution obtained with the chosen number of targets and can be improved by considering more acquisition points. However, the objective was to keep the mass of the targets negligible for all the cases considered, and for this reason, the number of acquisition points were kept constant throughout the experimental activity.

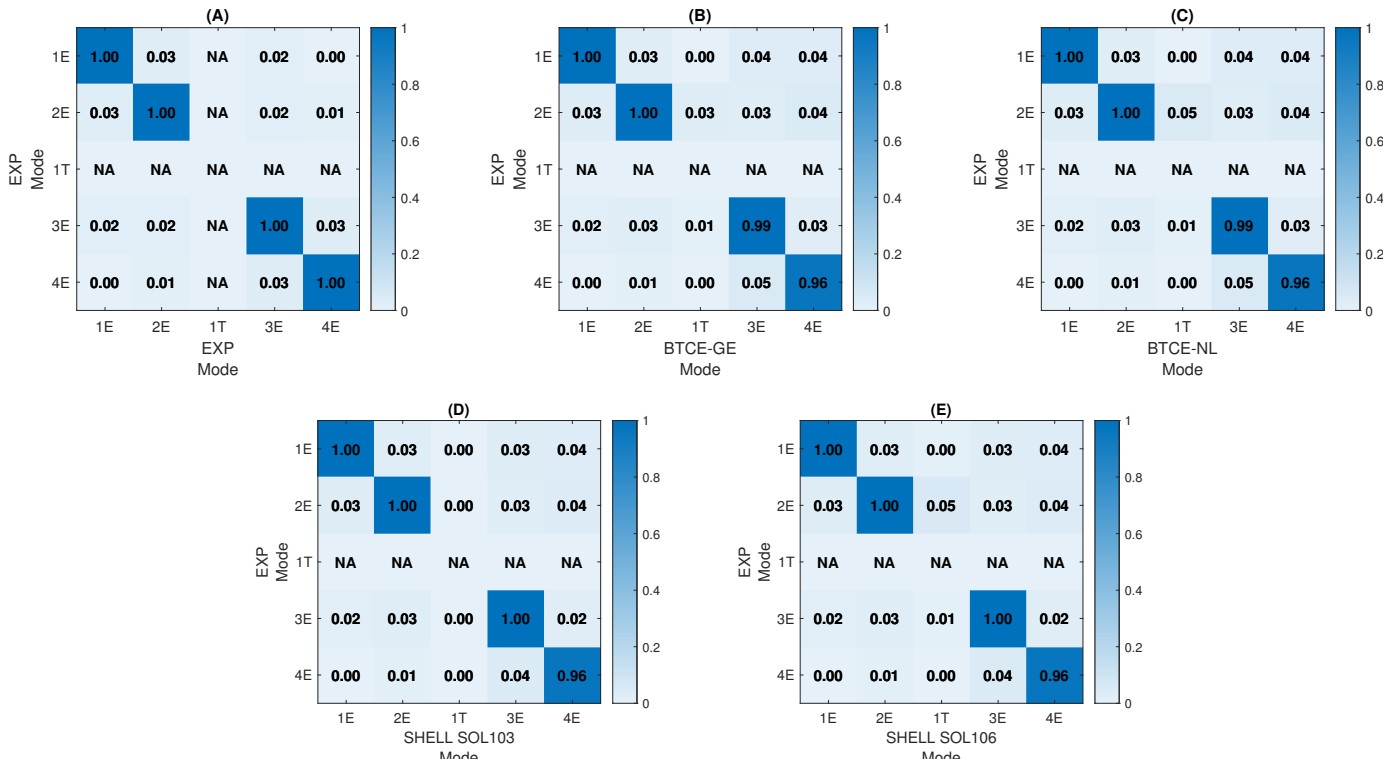

**Figure 9.** MAC $L$ = 1000 mm, comparison with Experimental Mode Shapes: Auto MAC (**A**), BTCE-GE (**B**), BTCE-NL (**C**), SHELL SOL 103 (**D**), SHELL SOL 106 (**E**).

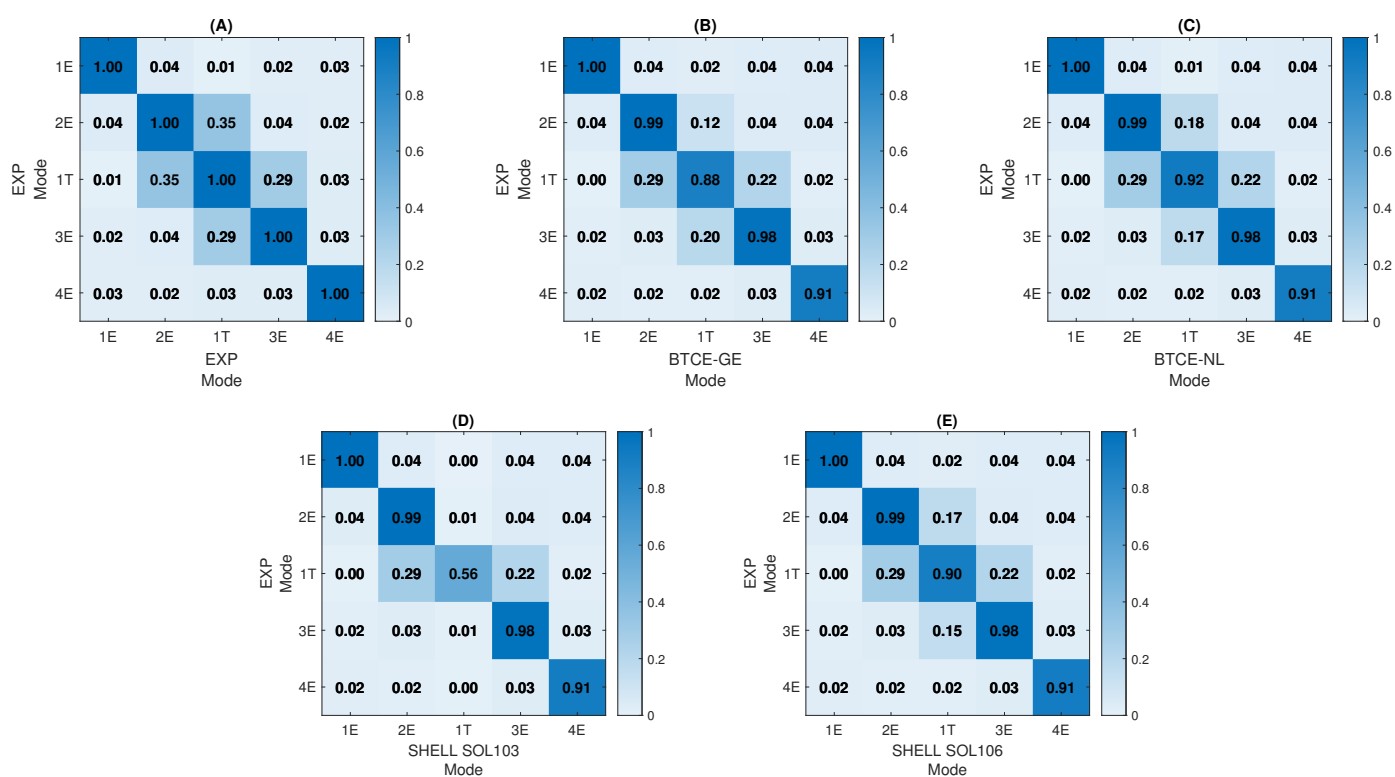

**Figure 10.** MAC $L$ = 1500 mm, comparison with Experimental Mode Shapes: Auto MAC (**A**), BTCE-GE (**B**), BTCE-NL (**C**), SHELL SOL 103 (**D**), SHELL SOL 106 (**E**).

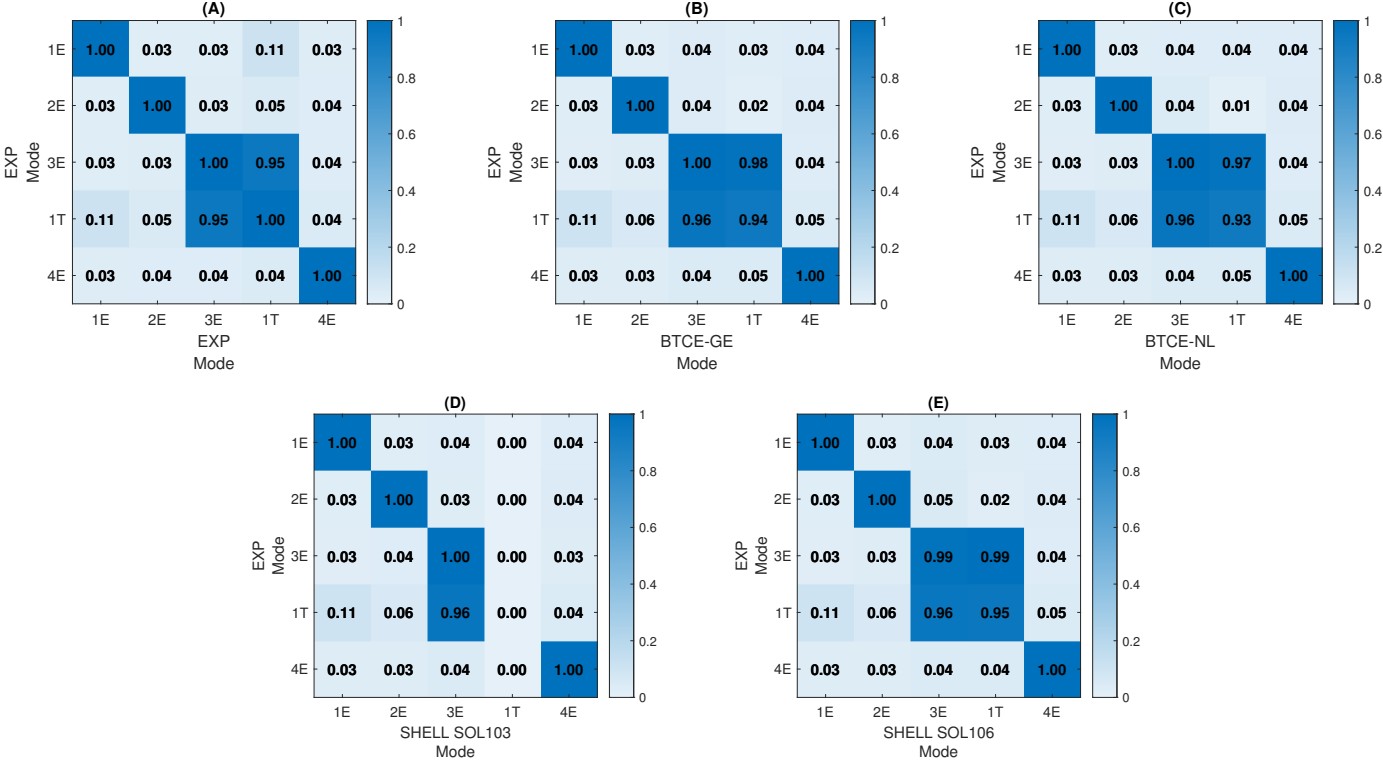

**Figure 11.** MAC $L$ = 2000 mm, comparison with Experimental Mode Shapes: Auto MAC (**A**), BTCE-GE (**B**), BTCE-NL (**C**), SHELL SOL 103 (**D**), SHELL SOL 106 (**E**).

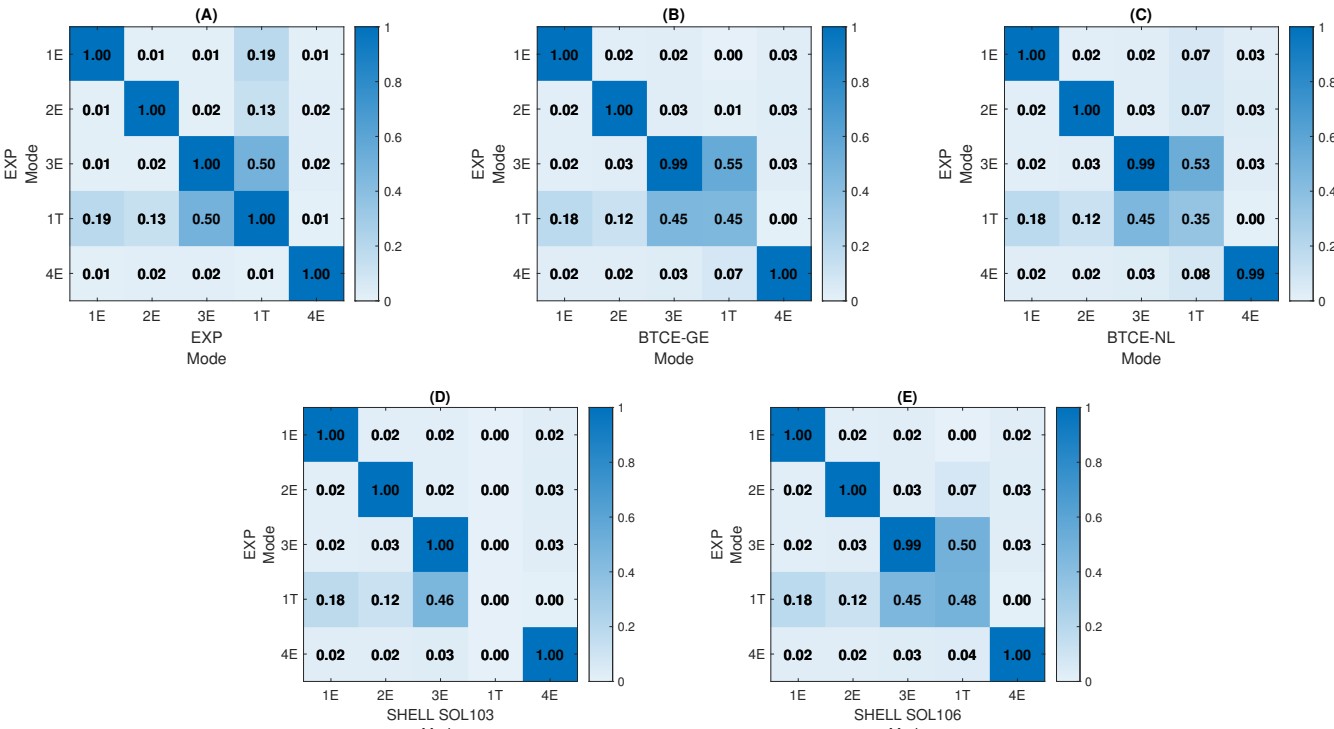

**Figure 12.** MAC $L$ = 2500 mm, comparison with Experimental Mode Shapes: Auto MAC (**A**), BTCE-GE (**B**), BTCE-NL (**C**), SHELL SOL 103 (**D**), SHELL SOL 106 (**E**).

*5.3. Numerical Modal Analysis Results*

The results of the numerical modal analysis of the composite box beam structure described by [37] are represented in Figures 13–16. Eight load cases were considered for a maximum deflection $\lambda = 28.55\%$; six of them correspond to a deflection below 10% and can help to observe more precisely at which point the nonlinear effects cause the deviation from the linear results. The results of the simulation performed with the BTCE models were compared to the frequencies obtained with a SHELL model solved with NASTRAN SOL106. For this comparison, the first eight modes were investigated. In this case, the comparison was performed on the frequencies computed with the nonlinear models normalized with their linear counterparts computed for the undeformed configuration. With this method, the variation of the characteristic frequency is highlighted. The material orientation causes the flapwise bending–torsion coupling, while the deflection causes the edgewise bending–torsion coupling; for this reason, all the modes involve three degrees of freedom. However, one component of the eigenvector has a considerably higher value than the other; for this reason, the modes where flapwise bending is the major effect are denoted with the letter F, while the modes where the edgewise bending component is predominant are denoted with the letter E, and the mainly torsional modes are denoted with the letter T, as done previously.

The results shows a good correlation between the BTCE models and the SHELL FE model. The first, second, fifth and seventh modes present very similar results, even for large displacements. The third, fourth, sixth and eighth modes present some discrepancies when the deformations are bigger than 15%. A less accurate prediction of the characteristic frequencies can be attributed to many factors. First of all, the number and the nature of the finite element used bring approximations that are necessary to lower the computational costs but can influence the results. Secondly, the hypothesis of inextensibility adopted for the BTCE could be unverified for large nonlinear deformations.Moreover, the curvatures and the rotation matrix are obtained under the hypothesis of moderate-to-large displacements. This comparison shows that the BTCE models could be used for the nonlinear

analysis of predeformed structures with deflection below 15% with results comparable to the characteristic modes of a SHELL FE model of the same structure solved with NASTRAN SOL106. Moreover, the results show that the differences between BTCE-GE and BTCE-NL are minimal, up to a deflection of 15% and increase for larger deformations. The models here presented can be further improved with an experimental test involving coupled structures to assess the performance and correctly evaluate the influence of geometrical and stiffness effects.

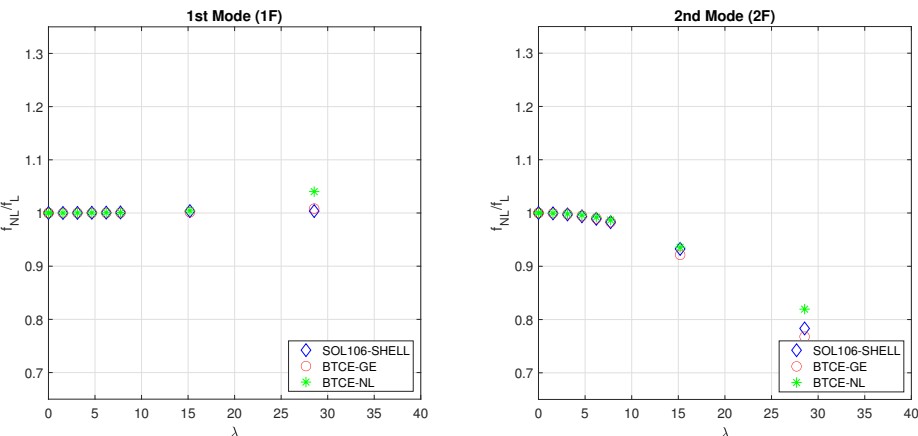

**Figure 13.** FE model results comparison for 1st and 2nd modes.

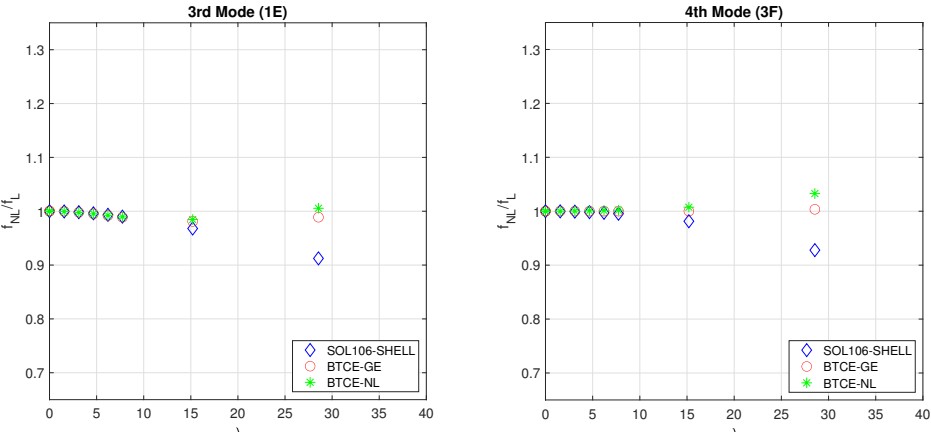

**Figure 14.** FE model results comparison for 3rd and 4th modes.

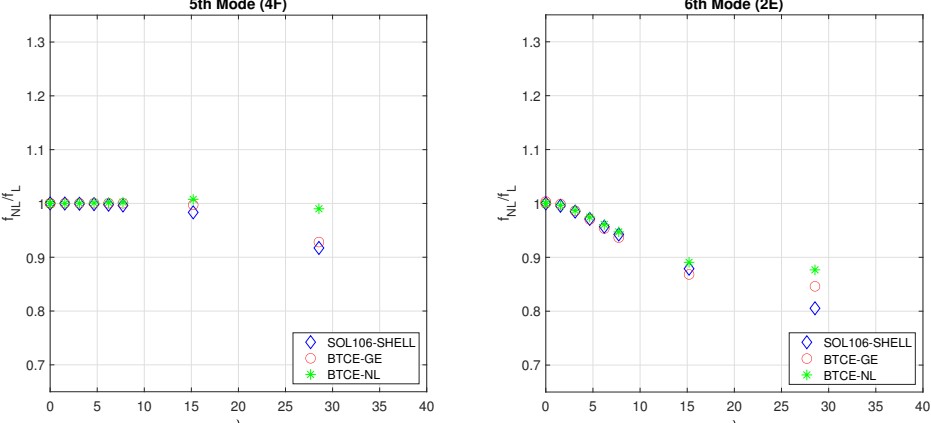

**Figure 15.** FE model results comparison for 5th and 6th modes.

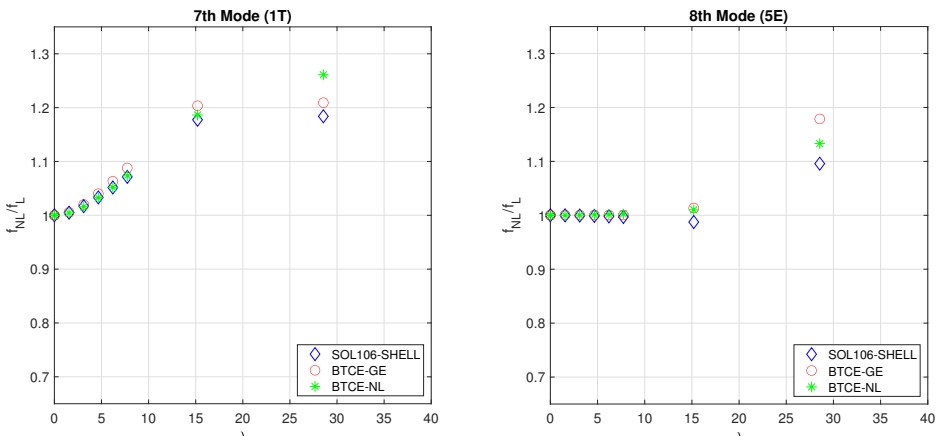

**Figure 16.** FE model results comparison for 7th and 8th modes.

## 6. Conclusions

This research work presented the derivation of two models for the dynamic analysis of beam structures with bending–torsion coupling in the presence of geometric nonlinearities. The first model accounts for the geometric effect through the orientation of the beam finite element according to a known equilibrium deformation. The second model accounts for nonlinear terms in the stiffness matrix derivation through the hypothesis of small perturbations of an equilibrium configuration under a static load. The stiffness matrix was derived with Hamilton's Principle. An experimental activity was carried out with the scope of verifying the level of deflection sufficient to have appreciable nonlinear effects and to assess the accuracy of the nonlinear analysis with the BTCE models. The experimental tests have been performed with a LDV system on an aluminum beam constrained at four different length, this allowed to study the nonlinear effects with four different level of deformation. The experiment showed that for the cases considered, the geometric nonlinearities have minor effects on the characteristic frequencies of the structure. Linear and nonlinear numerical models predicted frequencies generally within an error lower than 5%. Concerning the mode shapes, this research work revealed that for a deformation $\lambda$ below 3.7%, the mode shapes present a low level of coupling, and linear numerical models can be used to study structure under these conditions. On the other hand, for the cases where the deformation was 8.3% or 14.7%, the presence of nonlinear couplings determined relevant differences in the mode shapes, which were correctly predicted by the BTCE models here derived. Some of the results presented small differences between the observed modes and the ones predicted with the derived numerical models. These minor discrepancies are connected to the relatively low number of scanning point, which lowered the resolution. Moreover, the BTCE models rely on the equilibrium solution computed with a linear static analysis, which can be less accurate for higher deformations. The experimental activity showed that the stiffness effect plays a minor role for the analysis considered and that the BTCE-GE model can be sufficient for characteristic modes and frequencies prediction. The BTCE models were also tested for a composite structure with bending–torsion coupling. A numerical comparison revealed good accordance with the results obtained with a SHELL FE model solved with NASTRAN SOL106. The use of the presented model can be extended to the study of the aeroelastic performance of wing structures. Moreover, the bending–torsion coupling formulation allows to perform optimization on the material orientation to achieve desired dynamic properties also in the presence of geometrical nonlinearities.

**Author Contributions:** Conceptualization, C.P., E.C. and G.F.; methodology, C.P., E.C. and G.F.; validation, C.P.; formal analysis, C.P.; investigation, C.P., E.C. and G.F.; resources, E.C. and G.F.; data curation, C.P.; writing—original draft preparation, C.P., E.C. and G.F.; writing—review and editing, C.P., E.C. and G.F.; visualization, C.P.; supervision, E.C. and G.F. All authors have read and agreed to the published version of the manuscript.

**Funding:** This research received no external funding.

**Institutional Review Board Statement:** Not applicable.

**Informed Consent Statement:** Not applicable.

**Data Availability Statement:** The data presented in this study are available on request from the corresponding author.

**Acknowledgments:** The Authors wish to thank Simone Grendene for the technical support during the experimental activities.

**Conflicts of Interest:** The authors declare no conflicts of interest.

## Appendix A

**Table A1.** Shape functions for bending–torsion coupled beam element.

| $\{N_v(x)\}$ | | $\{N_w(x)\}$ | | $\{N_\varphi(x)\}$ | |
|---|---|---|---|---|---|
| $N_{v1}$ | $\left[1 - 3\dfrac{x^2}{L^2} + 2\dfrac{x^3}{L^3}\right]$ | $N_{w1}$ | $\left[1 - 3\dfrac{x^2}{L^2} + 2\dfrac{x^3}{L^3}\right]$ | $N_{\varphi 1}$ | $\left[1 - \dfrac{x}{L}\right]$ |
| $N_{v2}$ | $\left[x - 2\dfrac{x^2}{L} + \dfrac{x^3}{L^2}\right]$ | $N_{w2}$ | $-\left[x - 2\dfrac{x^2}{L} + \dfrac{x^3}{L^2}\right]$ | $N_{\varphi 2}$ | $\left[\dfrac{6K}{GJ_t L^3}(x^2 - Lx)\right]$ |
| $N_{v3}$ | $\left[3\dfrac{x^2}{L^2} - 2\dfrac{x^3}{L^3}\right]$ | $N_{w3}$ | $\left[3\dfrac{x^2}{L^2} - 2\dfrac{x^3}{L^3}\right]$ | $N_{\varphi 3}$ | $\left[\dfrac{3K}{GJ_t L^2}(Lx - x^2)\right]$ |
| $N_{v4}$ | $\left[-\dfrac{x^2}{L} + \dfrac{x^3}{L^2}\right]$ | $N_{w4}$ | $-\left[-\dfrac{x^2}{L} + \dfrac{x^3}{L^2}\right]$ | $N_{\varphi 4}$ | $\left[\dfrac{x}{L}\right]$ |
| | | | | $N_{\varphi 5}$ | $\left[\dfrac{6K}{GJ_t L^3}(Lx - x^2)\right]$ |
| | | | | $N_{\varphi 6}$ | $\left[\dfrac{3K}{GJ_t L^2}(Lx - x^2)\right]$ |

$$[K_{el}] = \begin{bmatrix} \dfrac{12EI_z}{L^3} & 0 & 0 & 0 & \dfrac{6EI_z}{L^2} & -\dfrac{12EI_z}{L^3} & 0 & 0 & 0 & \dfrac{6EI_z}{L^2} \\ & K_1 & 0 & K_2 & 0 & 0 & -K_1 & 0 & K_2 & 0 \\ & & \dfrac{GJ_t}{L} & \dfrac{K}{L} & 0 & 0 & 0 & -\dfrac{GJ_t}{L} & -\dfrac{K}{L} & 0 \\ & & & K_3 & 0 & 0 & -K_2 & -\dfrac{K}{L} & K_4 & 0 \\ & & & & \dfrac{4EI_z}{L} & -\dfrac{6EI_z}{L^2} & 0 & 0 & 0 & \dfrac{2EI_z}{L} \\ & & & & & \dfrac{12EI_z}{L^3} & 0 & 0 & 0 & -\dfrac{6EI_z}{L^2} \\ & & & & & & K_1 & 0 & -K_2 & 0 \\ & & & & & & & \dfrac{GJ_t}{L} & \dfrac{K}{L} & 0 \\ & & & & & & & & K_3 & 0 \\ & & & & & & & & & \dfrac{4EI_z}{L} \end{bmatrix} \tag{A1}$$

with

$$K_1 = \frac{12(EI_yGJ_t - K^2)}{GJ_tL^3} \quad K_2 = \frac{6(K^2 - EI_yGJ_t)}{GJ_tL^2}$$

$$K_3 = \frac{4EI_yGJ_t - 3K^2}{GJ_tL} \quad K_4 = \frac{2EI_yGJ_t - 3K^2}{GJ_tL}$$

(A2)

$$[M_{el}] = \begin{bmatrix} \frac{13AL\rho}{35} & 0 & 0 & 0 & \frac{11AL^2\rho}{210} & -\frac{9AL\rho}{70} & 0 & 0 & 0 & -\frac{13AL^2\rho}{420} \\ & M_1 & -\frac{I_pK\rho}{2GJ_t} & M_2 & 0 & 0 & M_3 & -\frac{I_pK\rho}{2GJ_t} & M_4 & 0 \\ & & \frac{I_pL\rho}{3} & \frac{I_pKL\rho}{4GJ_t} & 0 & 0 & \frac{I_pK\rho}{2GJ_t} & \frac{I_pL\rho}{6} & \frac{I_pKL\rho}{4GJ_t} & 0 \\ & & & M_5 & 0 & 0 & -M_4 & \frac{I_pKL\rho}{4GJ_t} & M_6 & 0 \\ & & & & \frac{AL^3\rho}{105} & \frac{13AL^2\rho}{420} & 0 & 0 & 0 & -\frac{AL^3\rho}{140} \\ & & & & & \frac{13AL\rho}{35} & 0 & 0 & 0 & -\frac{11AL^2\rho}{210} \\ & & & & & & M_1 & \frac{I_pK\rho}{2GJ_t} & -M_2 & 0 \\ & & & & & & & \frac{I_pL\rho}{3} & \frac{I_pKL\rho}{4GJ_t} & 0 \\ & & & & & & & & M_5 & 0 \\ & & & & & & & & & \frac{AL^3\rho}{105} \end{bmatrix}$$

(A3)

with

$$M_1 = \frac{6I_pK^2\rho}{5GJ_t^2L} + \frac{13AL\rho}{35} \quad M_2 = -\frac{3I_pK^2\rho}{5GJ_t^2} - \frac{11AL^2\rho}{210} \quad M_3 = -\frac{6I_pK^2\rho}{5GJ_t^2L} + \frac{9AL\rho}{70}$$

$$M_4 = -\frac{3I_pK^2\rho}{5GJ_t^2} + \frac{13AL^2\rho}{420} \quad M_5 = \frac{3I_pK^2L\rho}{10GJ_t^2} + \frac{AL^3\rho}{105} \quad M_6 = \frac{3I_pK^2L\rho}{10GJ_t^2} - \frac{AL^3\rho}{140}$$

(A4)

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
