# Peer review of "A Nonlinear Beam Finite Element with Bending–Torsion Coupling Formulation for Dynamic Analysis with Geometric Nonlinearities"

_aerospace, doi:10.3390/aerospace11040255_

Round 1
Reviewer 1 Report
Comments and Suggestions for Authors
Overall, interesting manuscript with valuable technical content. Some comments:
1. It appears that the experimental perturbations were not sufficient to markedly show a significant difference between linear and nonlinear responses. While differences do appear as the length of the beam and the displacement increase, would have been desirable to generate even larger amplitudes to truly showcase the necessity and importance of nonlinear FEM models.
2. The MAC figures are mislabeled, with figures going from A to E but the captions only going from A to D. This needs to be corrected.
3. On page 12, Line 303, instead of "5%", it is written as "%5"; on Page 19, Lines 405 and 407, should state "3.7%" instead of "3,7%", and "8.3%", "14.7%" instead of "8,3%" and "14,7%"
Comments on the Quality of English LanguageThere are some issues with use of English grammar, including conjugation and use of plural statements. For example:
1. Line 19, it is a run-on sentence, should be two separate sentences
2. Lines 27, 28: should read "These models can be very sophisticated and generally require a large number of calculations..."
3. Line 34: should read "These models..."
4. Line 46: should read "...and employ different solutions..."
5. Line 49: should read "...developed a six degree of freedom beam..."
6. Line 73: should read "...limited to composite materials, as several studies have shown that..."
7. Section 2 "Models Derivation" should read "Model Derivations"
8. There are many other similar English language grammar and corrections that need to be implemented before this paper can be considered for publication.
Author Response
Overall, interesting manuscript with valuable technical content. Some comments:
- It appears that the experimental perturbations were not sufficient to markedly show a significant difference between linear and nonlinear responses. While differences do appear as the length of the beam and the displacement increase, would have been desirable to generate even larger amplitudes to truly showcase the necessity and importance of nonlinear FEM models.
The reviewer highlights a key finding of this research work, the non-linear behaviour of the structure is highly affected by the entity of the deformation. The experiments on the aluminium bar revealed that the frequencies do not present significant modifications, but the effects on the mode shapes are present. Generate larger amplitudes would probably have a greater impact on the frequencies but the problem would be outside of the model hypothesis (moderate to large deflections). The numerical case with the composite beam showed greater differences in the frequencies even for moderate deformations, this shows that the effects of equilibrium deformations on characteristics frequencies depends also on the structure and on the couplings involved.
- The MAC figures are mislabeled, with figures going from A to E but the captions only going from A to D. This needs to be corrected.
Captions Corrected
- On page 12, Line 303, instead of "5%", it is written as "%5"; on Page 19, Lines 405 and 407, should state "3.7%" instead of "3,7%", and "8.3%", "14.7%" instead of "8,3%" and "14,7%"
Corrected
There are some issues with use of English grammar, including conjugation and use of plural statements. For example:
- Line 19, it is a run-on sentence, should be two separate sentences
- Lines 27, 28: should read "These models can be very sophisticated and generally require a large number of calculations..."
- Line 34: should read "These models..."
- Line 46: should read "...and employ different solutions..."
- Line 49: should read "...developed a six degree of freedom beam..."
- Line 73: should read "...limited to composite materials, as several studies have shown that..."
- Section 2 "Models Derivation" should read "Model Derivations"
- There are many other similar English language grammar and corrections that need to be implemented before this paper can be considered for publication.
The authors thank the reviewer for the corrections. The manuscript has been changed accordingly and carefully revised.
Reviewer 2 Report
Comments and Suggestions for Authors
Nonlinearity plays a pivotal role in analyzing aeroelastic effects within beam structures. Many of the existing studies mainly used linear beam theory, which greatly reduced the complexity by means of linear approximations, thus making it possible to analytically solve the strong-coupled. Yet, as the authors stated in their manuscript, such models may not always hold valid, particularly in the context of aeroelastic instabilities, such as flutter. The authors derived two models that incorporate geometric nonlinearities, one through the coordinate transformation of beam FE and the other via the introduction of nonlinear terms to stiffness matrix. Their derivations appear sound for me, but the authors should describe the derivation processes more clearly (see my detailed comments). Their experimental validation was certainly beneficial for this work, and the authors compared other existing numerical and analytical models with theirs. The comparison results demonstrated the advantages of BTCE-GE and BTCE-NL in the case of bending-torsion coupling. I would recommend the publication of this manuscript once my following concerns can be successfully addressed.
1. In lines 116-119, the reader would have a better understanding if the authors can provide the information about which DOFs (e.g., extension, in-plane bending, out-of-plane bending, and rotation) describe the dynamics of the beam. Please describe it with the help of Figure 1.
2. In line 124, I think it is necessary to specify that the CAS configuration ignores the axial extension.
3. In line 153, the alphanumeric representation of the Cartesian ordinate system xyz differs from that labeled in Figure 2 (XYZ).
4. Were ϕ0, w0, and v0 in Eq. (12) obtained from Eq. (3)? If so, please provide the information explicitly in the text.
5. In Table 3, indicate that µ is tip deflection and L is the beam length in the first row of the table.
Author Response
Nonlinearity plays a pivotal role in analyzing aeroelastic effects within beam structures. Many of the existing studies mainly used linear beam theory, which greatly reduced the complexity by means of linear approximations, thus making it possible to analytically solve the strong-coupled. Yet, as the authors stated in their manuscript, such models may not always hold valid, particularly in the context of aeroelastic instabilities, such as flutter. The authors derived two models that incorporate geometric nonlinearities, one through the coordinate transformation of beam FE and the other via the introduction of nonlinear terms to stiffness matrix. Their derivations appear sound for me, but the authors should describe the derivation processes more clearly (see my detailed comments). Their experimental validation was certainly beneficial for this work, and the authors compared other existing numerical and analytical models with theirs. The comparison results demonstrated the advantages of BTCE-GE and BTCE-NL in the case of bending-torsion coupling. I would recommend the publication of this manuscript once my following concerns can be successfully addressed.
- In lines 116-119, the reader would have a better understanding if the authors can provide the information about which DOFs (e.g., extension, in-plane bending, out-of-plane bending, and rotation) describe the dynamics of the beam. Please describe it with the help of Figure 1.
Thank you for helping us improve the quality of the manuscript, the dynamics of the beam is now described in the text according to your suggestions.
- In line 124, I think it is necessary to specify that the CAS configuration ignores the axial extension.
The inextensibility of the beam is now specified in the text.
- In line 153, the alphanumeric representation of the Cartesian ordinate system xyzdiffers from that labeled in Figure 2 (XYZ).
The figure and the text are now consistent.
- Were ϕ0, w0, andv0 in Eq. (12) obtained from Eq. (3)? If so, please provide the information explicitly in the text.
Equation 3 presents the equilibrium deformation expressed in terms of shape functions and nodal degrees of freedom. While Equation 12 introduce the definition of the displacement variables as the sum of an equilibrium term and a perturbation term. The equilibrium term of Equation 12 can be expressed as in Equation 3, this substitution is performed in Equation 24 to solve the integral.
- In Table 3, indicate that µis tip deflection and L is the beam length in the first row of the table.
The variables have been defined in the caption of the table.